# A Community-Based Mixed-Methods Study: Fish Bycatch Protein Supplementation as a Sustainable Solution for Child Malnutrition in Bengaluru, India

**DOI:** 10.3390/nu17111751

**Published:** 2025-05-22

**Authors:** Kristen P. Yang, Sunil K. Khanna, Angela Chaudhuri, Syama B. Syam, Tammy M. Bray

**Affiliations:** 1School of Nutrition and Public Health, College of Health, Oregon State University, Corvallis, OR 97331, USA; sunil.khanna@oregonstate.edu (S.K.K.); tammy.bray@oregonstate.edu (T.M.B.); 2Swasti, Bengaluru 560094, Karnataka, India; angela@catalysts.org (A.C.); syama@catalysts.org (S.B.S.)

**Keywords:** child malnutrition, bycatch-derived protein supplementation, scalable sustainable nutrition, child growth and development, community-based intervention

## Abstract

**Objective:** Malnutrition remains a global challenge to child development, with urban slums in India experiencing high rates of protein deficiency. This study aimed to evaluate the effectiveness of a low-cost, fish bycatch-derived protein supplement in supporting catch-up growth among malnourished children. **Methods:** Using the Sustainable Community Partnership and Empowerment (SCOPE) model, we implemented a 90-day intervention with daily 10 g of Advanced Protein Powder (APP), produced from underutilized fish bycatch. Forty-six malnourished children (aged 3–6) from Bengaluru slums were randomized into a Control group receiving caloric support or an APP supplement group. Growth indicators, cognition, and caregiver perspectives were assessed. **Results:** Children receiving the APP supplement showed a significant increase in the weight-for-age percentile (underweight), rising by 7.59%, compared to 0.59% in the Control group (*p* = 0.02185). Muscle growth, measured by mid-upper arm circumference, also improved significantly in the APP Group (*p* < 0.05). In the first month, APP supplementation led to a significant height gain of 1.86 cm (*p* < 0.001), whereas the Control group showed no change (*p* > 0.05). Additionally, APP supplementation enhanced cognitive function, visual processing, short-term memory, and planning ability, with sustained effects at six months (*p* < 0.05). Caregivers reported noticeable improvements in children’s vitality, appetite, focus, and engagement in social and learning activities. **Conclusions:** Bycatch-derived protein supplementation, implemented through the SCOPE model, enhanced physical growth, behavior, and cognition in malnourished children in urban slums. Future studies should investigate the long-term effects, scalability, and adaptability of this sustainable solution for addressing child malnutrition.

## 1. Introduction

As the 2030 Sustainable Development Goals deadline approaches, child malnutrition remains a critical global public health challenge, with profound implications for socio-economic development [1]. Early-life malnutrition adversely affects linear growth, increases susceptibility to infections, and contributes to long-term health risks, cognitive impairment, and reduced productivity in adulthood [2]. India continues to have one of the highest burdens of child malnutrition despite extensive interventions led by local governments and global organizations [3]. National data indicate that malnutrition rates remain alarmingly high, with 36% of children under five experiencing stunting, 19% wasting, and 32% being underweight [4]. Urban slums, where poor sanitation, poverty, and food insecurity converge, continue to face persistently high rates of malnutrition [5]. Particularly in slums of Mumbai, limited child-feeding practices, food insecurity, and substandard living conditions have been directly linked to undernutrition [6]. Furthermore, slum-based studies in Bangladesh found poor uptake and acceptability of ready-to-use therapeutic foods due to community resistance and lack of engagement, highlighting the need for culturally appropriate interventions [7]. In Bengaluru, a rapidly growing Indian metropolis where 22.56% of the population live in 387 informal settlements, community-based screening revealed that 41% of children under five were underweight and 48.5% were stunted [8]. Thus, these findings underscore the urgent need for context-specific, culturally sensitive, and community-driven interventions tailored to the challenges of child malnutrition in urban slums in Bengaluru, India.

Among nutritional strategies, ensuring adequate intake of protein and essential amino acids from animal-source foods is essential for promoting linear growth and development among malnourished children in early childhood [9]. Correspondingly, Indian national data show that complementary feeding of children with animal-sourced foods is associated with lower malnutrition rates [10]. Yet, cultural traditions, diverse religious practices, local food availability, environmental concerns, and limited-protein, staple-based food policies have limited access to animal-source foods and exacerbated protein deficiency among malnourished children in India [11,12]. Researchers have recently delved into alternative protein sources, including crickets, sago worms, breadfruit, and palm weevil larvae, which are considered promising for addressing child malnutrition in resource-limited settings but often face consumer resistance, particularly among children [13,14,15]. In contrast, fish protein, widely accepted and nutritionally rich, has demonstrated effectiveness in reducing child stunting [16,17]. One study suggests that fish-based interventions, including locally produced dried fish powder, offer cost-effective and accessible solutions for protein-deficient malnutrition [18]. Additionally, repurposing fish bycatch and byproducts aligns with sustainability efforts while addressing food insecurity [19,20]. Therefore, utilizing high-quality protein derived from fish bycatch and byproducts presents a sustainable, affordable, and culturally acceptable strategy to improve linear growth and address child malnutrition in urban slums.

Building on these findings, this study investigates the effectiveness of Ocean-Based Advanced Protein Powder (APP), derived from fish bycatch, as a sustainable, high-quality, and affordable source of protein for malnourished children in urban slums in India. A prior laboratory study confirmed the safety and efficacy of APP supplementation in supporting weight gain and lean muscle development in the protein-malnourished young mouse model [21,22]. Drawing on this scientific foundation, the present study adopts a community-based and culturally appropriate approach by implementing APP supplementation within the Sustainable Community Partnership and Empowerment (SCOPE) framework, in collaboration with local non-profits, the food industry, and childcare facilities [23,24]. The research team hypothesized that a 90-day APP supplementation regimen would improve growth outcomes in malnourished children aged 3 to 6, a critical period for physical and cognitive development. By incorporating APP into culturally familiar dishes and assessing acceptability among children and caregivers, this study provides mixed-methods evidence supporting APP as an effective, sustainable, and community-driven nutritional intervention. Furthermore, it establishes a scalable model that integrates sustainable protein supplementation and the SCOPE framework to address child malnutrition in urban slums and lays the groundwork for future investigation to evaluate the long-term impacts on child growth and adaptability of sustainable protein support across diverse geographic and cultural settings worldwide.

## 2. Materials and Methods

### 2.1. Participants

This study initially recruited 100 children aged 3–6 years and their primary caregivers living in the urban slums of Bommanahalli, Bengaluru, India, from November 2021 to December 2022, representing approximately 24.87% of the 402 households in the targeted community during the preliminary screening. Conducted in collaboration with Swasti, a non-profit organization supporting marginalized communities and low-income families, these child–parent participants were selected from the i4We program, Swasti’s integrated healthcare framework [25]. Although the sampling strategy limited broader generalizability, the clearly defined and demographically concentrated setting enhances the validity of the findings for similar high-risk urban contexts.

After four participants withdrew, 96 children remained at the study’s outset, all below the 50th percentile for growth per WHO and Indian Academy of Pediatrics standards [26]. Eligible children were from low-income urban slum households and free from congenital, metabolic, or chronic diseases. Informed consent was obtained, and the documents were translated into Kannada. The Institutional Review Boards of Oregon State University (Corvallis, OR, USA; IRB-2019-0330, approved on 21 November 2019) and the Catalyst Foundation (Bengaluru, India; approved on 26 August 2019) approved the study protocol. Due to COVID-19 lockdowns and household challenges, 50 participants dropped out during the intervention. No adverse events were reported. The final sample included 19 participants in the Control group and 27 in the APP group, representing 11.44% of total household in the target population. The study flow is presented in Figure 1.

### 2.2. The Feeding Approach of APP Supplementation and Control Diets

Researchers conducted a 90-day nutritional intervention with a follow-up assessment on Day 180 to evaluate the effects of APP supplementation on malnourished children. Participants were randomly assigned to either the Control group (*n* = 47), receiving a standard 450–500 Kcal meal, or the APP group (*n* = 49), receiving the same meal supplemented with 10 g of APP. Baseline data indicated an average daily energy intake of 908 Kcal, a 33% deficit from the recommended dietary allowance (RDA) of 1360 Kcal [27]. The intervention aimed to bridge this gap by providing one-third of the daily energy requirement through culturally appropriate meals. Initially, 20 g of APP per meal was tested, but this was adjusted to 10 g based on better acceptance among caregivers and children [28]. Nutritional compositions of APP and experimental diets are detailed in Appendix A and Appendix B.

Within the Sustainable Community Partnership and Empowerment (SCOPE) model framework, the feeding intervention was designed to be culturally sensitive, child-centered, and community-driven. In partnership with local stakeholders, including community organizations, caregivers, and certified dietitians, meals were developed to align with traditional food preferences while optimizing nutrient density. Standardized recipes incorporated at least 10 g of vegetables per meal, reinforcing the importance of dietary diversity. The cyclic menu featured culturally familiar dishes such as tomato rice, vegetable pulao, potato patties, and date and puffed rice bars, enhancing community engagement and food security while maintaining nutritional adequacy.

Meals were prepared under strict hygiene protocols at the i4We community center kitchen, adhering to COVID-19 safety measures to ensure food safety and public health compliance. Children dined in a structured group setting, fostering social interaction, peer support, and positive eating behaviors. A reward system was introduced to encourage program adherence and participation, incentivizing daily attendance and meal completion. Researchers also monitored food intake and meal acceptance by tracking leftovers, allowing for continuous program refinement. This structured yet adaptable feeding approach addressed critical protein deficits and reinforced community-led solutions, emphasizing the long-term sustainability of child nutrition programs in resource-limited settings.

### 2.3. Indicators of Growth and Malnutrition Levels

Growth indicators were assessed using anthropometric measurements, including height, weight, mid-upper arm circumference (MUAC), bone mass density, and hemoglobin levels. Measurements were conducted using a digital weight scale (Tanita HD 382, Tokyo, Japan), a stadiometer (Maa International, New Delhi, India), and a measuring tape from a local health facility. Bone mass density was evaluated using the Ultrasound Bone Densitometer CM-300 (Furuno Electric Co., Ltd., Hyogo, Japan) at the calcaneus (heel bone). Hemoglobin levels were measured with the Hemoglobin Analyzer Hematology System (HemoCue HB 201+ 121721-EW11, Ängelholm, Sweden) by a registered nurse at the i4We community facility, following strict safety protocols for finger-prick sample collection. Anemia levels were classified according to the Indian National Health Survey criteria [4].

Malnutrition was assessed using Z-scores and percentiles for weight-for-age and height-for-age, based on WHO Child Growth Standards and local guidelines from the Indian Academy of Pediatrics and the Health Promotion Administration [26]. Malnutrition severity was categorized as severe (Z-score ≤ −3), moderate (−3 < Z-score ≤ −2), mild (−2 < Z-score ≤ −1), at-risk (−1 < Z-score < 0), and non-underweight/stunted (Z-score ≥ 0) [29]. These classifications allowed researchers to evaluate the impact of APP supplementation on growth outcomes across varying degrees of stunting (low height for age) and underweight (low weight for age).

### 2.4. Cognition Assessment

Cognitive function was evaluated using the Kaufman Cognitive Assessment Battery for Children (KCAB-II) (Pearson, Bloomington, MN, USA), based on Luria’s neurophysiological model and the Cattell–Horn–Carroll theory of cognitive abilities [30]. The assessment included nonverbal tasks to accommodate participants’ limited English proficiency. The test measured multiple cognitive domains: face recognition, where children identified correct faces from briefly displayed images; pattern reasoning, assessing logical pattern completion; and conceptual thinking, which required selecting the image that did not belong. Additionally, visual processing examined the ability to perceive and manipulate patterns, while hand movement assessed short-term memory through sequential tap replication. The results were compared against global norms, including benchmarks for Asian children aged 3–6 years.

### 2.5. Dietary Intake Assessment

Dietary intake was evaluated through five 24 h recalls, assessing macronutrients and micronutrient consumption, including daily energy (Kcal/day) and protein (g/day), by the 2020 Indian dietary reference intakes (DRIs), which encompass the recommended dietary allowance (RDA) and the estimated average requirement (EAR) [27].

The assessment covered all meals and snacks consumed by children throughout the day, including their habitual diets and intervention meals. Nutrient analysis was conducted using the Indian government’s food composition database and processed with DietSoft software (IDeAS, Noida, Uttar Pradesh, India) [31]. Trained workers, under the supervision of a certified dietitian, collected intake data using standardized containers for accurate dietary measurement. Caregivers provided ingredient details for mixed dishes, and researchers verified nutritional labels from local markets. All dietary data were reviewed and analyzed by a registered dietitian and local nutritionists, with nutrient intake compared to Indian DRIs for children aged 3–6 years to evaluate dietary adequacy and the impact of supplementation.

### 2.6. Statistics Analysis

All statistical analyses were conducted using R software (Version 4.4.0, The R Foundation, Vienna, Austria). Due to the small sample size, data were expressed as mean ± standard error of the mean (SEM). The Shapiro–Wilk test confirmed a nonparametric data distribution. Group differences were analyzed using the Wilcoxon rank sum test, while within-group differences across time points were assessed using the Friedman test, followed by post hoc Wilcoxon tests. Statistical significance was set at *p* < 0.05.

### 2.7. Focus Group Discussions with Child Caregivers

Eight focus group discussions (FGDs) were conducted to explore caregivers’ perspectives on protein supplementation and intervention strategies for malnourished children in Month 3 and Month 9. Each FGD included six caregivers with direct experience in feeding malnourished children. Using a semi-structured guide, discussions covered observations, perceptions, and challenges related to protein supplementation. Sessions lasted approximately forty minutes and were audio-recorded with the participants’ consent. Data were analyzed using a grounded theory approach, allowing themes to emerge organically. Transcripts were translated from Kannada to English by frontline health practitioners and analyzed using NVivo 12 software (QSR International Ltd., Melbourne, Australia) through open, axial, and selective coding to identify key patterns and construct a theoretical model of caregivers’ attitudes and practices.

## 3. Results

### 3.1. Baseline Measurement and Dietary Intervention

Table 1 summarizes the baseline growth indicators and dietary intake of the malnourished children. The mean age was 4.31 years, with no significant differences between the APP and Control groups in anthropometric parameters, including body weight, weight-for-age percentile (WAP), underweight prevalence, height, height-for-age percentile (HAP), stunting, and mid-upper arm circumference (MUAC) (*p* > 0.05). The average weight was 14.77 ± 0.61 kg, below the national standard of 18.3 kg, with a mean WAP of 24.07 ± 4.1. The mean height was 97.55 ± 1.41 cm, with an HAP of 19.82 ± 3.58, indicating more pronounced stunting than being underweight. The APP group had significantly higher bone mineral density (BMD) than the Control group (*p* = 0.03903), though both remained within the normal range. Mean hemoglobin levels (9.17 g/dL) indicated moderate anemia in both groups.

Baseline dietary intake, including energy, protein, fat, and carbohydrates, showed no significant differences between groups (*p* > 0.05). Both groups had an energy deficit of approximately 450 Kcal/day. During the intervention, energy intake increased in both groups, with no significant difference (*p* = 0.2139). Fat and carbohydrate intake remained comparable (*p* > 0.05), but the APP group consistently had significantly higher protein intake (*p* < 0.001), averaging an increase of 14.99 ± 1.57 g/day compared to 7.33 ± 1.69 g/day in the Control group (*p* = 0.001894). These findings suggest that providing one hot meal daily, alongside regular dietary intake, helped meet energy requirements in both groups while significantly enhancing protein intake in the APP group.

### 3.2. Impact of APP Supplementation Diets on Body Weight

Table 2 presents the body weight measurements at baseline, Month 1, Month 2, Month 3 (the end of the intervention), and Month 9 (follow-up). Both groups showed significant weight increases over the three-month intervention and follow-up (*p* < 0.001). However, no significant differences were observed between groups at any point (*p* > 0.05). Neither group reached the RDA-recommended weight of 18.3 kg during the study.

Participants were categorized as non-underweight, risky, mild, or moderate + severe underweight to assess the nutritional impact better. No significant weight changes were observed in the non-underweight category for the APP (*p* = 0.3007) or the Control group (*p* = 0.406). However, in the risky, mild, and moderate + severe groups, the APP group demonstrated significant weight gains at each time point compared to baseline (*p* < 0.05). In contrast, the Control group showed no significant changes during the three-month intervention (*p* > 0.05). These findings suggest that APP supplementation effectively promotes weight gain in underweight children, particularly those at higher risk of malnutrition.

### 3.3. Impact of APP Supplementation Diets on Height

Table 3 presents height measurements over the study period. Both groups exhibited significant height increases over time (*p* < 0.01), with no significant differences between them at any time point (*p* > 0.05). However, compared with the baseline, the APP group exhibited a significant height increase, averagely 1.86 cm as early as Month 1 (*p* < 0.001). In contrast, the Control group’s growth was delayed until Month 2. After the 90-day intervention, no significant height changes from baseline were observed in the Control group across all subcategories of stunting (*p* > 0.05). In contrast, the APP group demonstrated significant height increases in the risky, mild, and moderate + severe stunting subgroups at Months 1, 2, 3, and follow-up compared to baseline (*p* < 0.05). These findings suggest that APP supplementation may be particularly beneficial for promoting height growth in stunted children.

### 3.4. Impact of APP Supplementation Diets on Underweight and Stunting Improvement

After the three-month intervention, the weight-for-age percentile (WAP), an indicator of underweight malnutrition, increased significantly in the APP group (7.59 ± 2.48%) compared to the Control group (0.59 ± 1.57%, *p* = 0.02185). Figure 2a illustrates that the APP group showed significant WAP improvements across all underweight categories (*p* = 0.03006, 0.005952, and 0.02928 for riskily, mildly, and moderately + severely underweight, respectively). In contrast, the Control group only significantly improved in the mildly underweight category (*p* = 0.01971). Notably, in the moderately + severely underweight subgroup, WAP in the APP group increased from 0.83 ± 0.23 at baseline to 3.14 ± 1.03 at Month 1, significantly surpassing the Control group’s WAP of 0.18 ± 0.18 (*p* = 0.02094).

At baseline, 29.63% of children in the APP group and 21.05% in the Control group were moderately to severely underweight. By the end of the intervention, 50% of malnourished children in the APP group showed recovery, while no recovery was observed in the Control group. The adjusted odds ratio for weight improvement in the APP group was 9 (*p* = 0.208), suggesting a potential association between APP supplementation and recovery. However, further studies with larger samples are needed to confirm these findings.

The height-for-age percentile (HAP), an indicator of stunting severity, increased significantly over time in both groups as part of normal growth (*p* < 0.001), with no significant differences between them (*p* > 0.05). Figure 2b shows that within stunting subgroups, the APP group exhibited significant HAP improvements across all categories (*p* = 0.03693, 0.0001198, 0.04729). In contrast, the Control group showed significant improvement only in the moderate + severe category (*p* = 0.008487). Therefore, these findings suggest that while APP supplementation supports weight gain and may promote height growth, larger-scale interventions, and extended study durations are necessary for a more comprehensive impact on stunting.

### 3.5. Impacts of APP Supplementation on Bone Mass Density, Muscle Mass Gain, and Hemoglobin

Bone mass density (BMD) was assessed at baseline, Month 3, and Month 9 in both groups. The APP group exhibited more stable BMD values (0.89 ± 0.14, −0.004 ± 0.39, −0.5 ± 0.23) compared to the Control group (0.26 ± 0.25, −1.22 ± 0.2, −0.42 ± 0.2). As shown in Figure 3a, BMD was significantly higher in the APP group at both baseline (*p* = 0.03903) and Month 3 (*p* = 0.007924); however, by Month 9, the difference between groups was no longer significant (*p* = 0.3318). The Control group experienced a sharper decline in BMD, progressing toward osteopenia within the first three months, followed by partial recovery at follow-up. These findings suggest that APP supplementation may play a protective role in maintaining bone health. Further longitudinal studies using dual X-ray absorptiometry are needed to confirm these results.

Mid-upper arm circumference (MUAC), a key indicator of muscle mass gain, improved significantly in the APP group throughout the study (*p* < 0.05), as shown in Figure 3b. The APP group’s MUAC increased from 16.03 ± 0.48 cm (Baseline) to 16.45 ± 0.4 cm (Month 3) to 16.8 ± 0.44 cm (Month 9). In comparison, the Control group’s values increased from 15.88 ± 0.27 cm to 16.12 ± 0.31 cm to 16.24 ± 0.43 cm. The Control group showed a significant increase at Month 9 (*p* = 0.001366). In contrast, the APP group exhibited consistent, significant improvements throughout the intervention (*p* < 0.05). These results suggest that APP supplementation is critical in promoting muscle growth among malnourished children.

Hemoglobin (Hb) levels were assessed at baseline, Month 3, and Month 9, as shown in Figure 3c. The APP group’s Hb levels increased from 9.09 ± 0.51 g/dL (Baseline) to 10.39 ± 0.38 g/dL (Month 3) to 10.8 ± 0.44 g/dL (Month 9). Similarly, the Control group’s Hb levels improved from 9.08 ± 0.41 g/dL to 10.71 ± 0.2 g/dL to 10.99 ± 0.31 g/dL. Both groups demonstrated statistically significant Hb improvements (*p* < 0.05), transitioning participants from moderate anemia (Hb 7–9.99 g/dL) to mild anemia (Hb 10–10.99 g/dL). These findings suggest that protein-rich (APP) and energy-only (Control) feeding approaches effectively reduce anemia severity. APP supplementation performed better on key bone and muscle growth indicators, including BMD and MUAC, while also contributing to anemia reduction. These findings highlight its potential as a sustainable and effective intervention for addressing malnutrition.

### 3.6. Impacts on APP Supplementation on Cognition

Cognitive function was assessed using multiple tests, including conceptual thinking, visual processing, short-term memory, and planning ability (Figure 4). Within the APP group, all tests, except for conceptual thinking, showed significant improvements at different time points throughout the study (*p* < 0.05), whereas no significant changes were observed in the Control group (*p* > 0.05). Notably, these cognitive improvements persisted beyond the supplementation period. By Month 9, the APP group demonstrated significant improvements in the following assessments: triangle test (*p* = 0.007882)—evaluating visual processing; face recognition test (*p* = 0.003496)—assessing short-term memory; hand movement test (*p* = 0.04798)—measuring motor planning ability, and pattern reasoning test (*p* = 0.004798)—assessing logical pattern recognition.

As indicated in Figure 4, the global average scores for these cognitive tests were as follows: conceptual thinking 9.9; triangle test 9.5; hand movement test 9.8; and face recognition test 9.9. At Month 9, the APP group’s cognitive scores were either above or closer to the global average compared to the Control group, suggesting a potential long-term effect of APP supplementation on cognitive function in malnourished children. These findings suggest that APP supplementation may enhance cognitive abilities, particularly in visual processing, short-term memory, and planning ability. Further studies are necessary to explore the impact of early supplementation and the sustainability of these cognitive benefits.

### 3.7. Caregiver’s Perspectives on the APP Supplementation and the SCOPE Approach

The qualitative findings from the protein supplementation intervention revealed notable improvements in the physical growth, eating behaviors, and cognitive development of malnourished children.

#### 3.7.1. Physical Growth

Caregivers consistently observed visible improvements in their children’s height and weight, attributing these changes to the nutrient-rich meals, particularly protein supplementation provided at the community center. Many parents expressed satisfaction with their children’s growth, highlighting the noticeable impact of the intervention. One caregiver remarked:


*“Facilities are good. Though we have not tasted the food yet, by the way the children are enjoying their meals, we can say that it is delicious. We have seen some positive changes in our children. They are growing. Height and weight are also improved. We are happy that the children have liked this food.”*


Another caregiver emphasized the remarkable transformation in their child’s physical development from their family’s observation:


*“My husband has observed a lot of changes in our daughter. Initially, she was very weak. Now, everyone is surprised by her height and weight. Even the others have the same opinion about their children. Changes are observable and positive.*


These testimonials underscore the effectiveness of the intervention in addressing malnutrition and supporting healthy growth among children in resource-limited settings.

#### 3.7.2. Eating Behavior Change

The culturally appropriate, child-centered approach with nutritional education implemented through the SCOPE model significantly impacted children’s eating behaviors. Many children who previously exhibited poor eating habits or were picky eaters began consuming a wider variety of foods, particularly vegetables, due to the appealing, well-prepared meals at the center. One mother observed:


*“If we give vegetables to our children at home, they won’t eat them. Here they are mixing carrots, potatoes, and tomatoes with food and boiling them. They simply eat here. If we give them vegetables at home, they just put them aside.”*


The group dining environment enhanced appetite and willingness to try new foods. Eating alongside peers fostered a sense of community, encouraged social interaction, and motivated children to eat better. One parent emphasized:


*“This place and the friends around give them a fresh air of change. Many times, my children were not able to eat, but this atmosphere has made it possible for them to have qualitative and nutritious food. I am happy to see them play here and eat well.”*


The structured and hygienic environment at the center further enhanced the eating experience. Parents also noted positive changes in eating habits at home, as children demonstrated greater self-regulation and willingness to try new foods. One caregiver observed:


*“As we commenced our visits here, my son started to eat on his own. Earlier, he was using his mobile phone too much, but now he is engaged in activities and eats without distractions.”*


Children were also motivated by reward systems, such as earning stars for finishing their meals, encouraging them to eat more. A parent explained:


*“Children are given stars for finishing their meals. This is a motivating factor, which makes them fill their plates again.”*


These findings highlight the effectiveness of a structured, socially engaging, and culturally relevant approach in improving dietary habits and promoting a more diverse and nutritious diet among malnourished children.

#### 3.7.3. Cognitive Improvements

Beyond physical improvements, caregivers reported remarkable enhancements in cognitive function. Children demonstrated better concentration, self-confidence, and academic performance, with improvements extending to problem-solving skills. One caregiver shared:


*“Yes, a lot of changes have happened in her reading, which makes me happy. She is so brilliant now. Even her teacher has observed these changes.”*


Another parent highlighted their child’s newfound independence and decision-making skills:


*“Now my daughter understands almost everything and does her homework on her own. She even helps me with my routine and is able to take decisions responsibly.”*


These caregiver testimonials illustrate that the fish waste protein intervention had multifaceted benefits, improving not only malnourished children’s height and weight but also cognitive abilities, eating habits, and social behaviors. The findings underscore the holistic impact of the nutritional support program, demonstrating its effectiveness in addressing malnutrition while fostering overall developmental and behavioral growth in children.

## 4. Discussion

This community-based study uniquely demonstrates that supplementing with ocean-based protein (APP), derived from fish bycatch, is an effective strategy for enhancing physical and cognitive growth among malnourished children in Indian slums. APP is sustainably extracted from underutilized, food-grade wild-caught ocean fish, offering a low-carbon and land- and water-efficient alternative to traditional protein [32]. A previous animal study confirmed the safety and effectiveness of APP supplementation in promoting weight gain and muscle growth in protein-deficient young mice [22]. This study further extends those findings to a human population through the Sustainable Community Partnership and Empowerment (SCOPE) model, which enabled culturally appropriate, community-driven implementation via local NGOs, the food industry, and childcare providers. Despite initial concerns about fish flavor and perceptions of waste observed in other studies, these caregivers widely recognized fish growth-enhancing benefits of fish protein and reported high acceptance [33]. Ragsdale et al. (2024) similarly found that dried fish protein powder was highly accepted among Zambian children, effectively addressing nutritional gaps [34]. The mixed-methods findings of this study position the fish protein powder, embedded with the SCOPE framework, as a viable, culturally appropriate, and sustainable solution to address moderate-to-severe malnutrition and protein insecurity among children in low-resource urban settings.

Studies show that protein supplementation for malnourished children enhances growth and development [35,36]. Addressing malnutrition in low-resource settings requires high-quality protein with essential amino acids crucial for child development [37,38]. Scholars emphasize protein quality, making APP’s complete protein profile, rich in methionine, cysteine, phenylalanine, tryptophan, valine, and lysine, a superior nutritional source to address amino acid deficiency and protein malnutrition [39]. A recent review of interventions in urban slums across Africa, Asia, and Central America, highlighted a focus on micronutrients and nutrition education, with limited emphasis on high-quality protein supplementation for malnourished children [40]. Recognizing the critical role of protein in child growth, this study shows that APP supplementation significantly improved energy and protein intake, leading to notable gains in weight and height, aligning with Indian RDA guidelines and reducing severe malnutrition in urban slums [27]. Additionally, qualitative findings from caregivers highlighted noticeable improvements in children’s energy levels, appetite, focus, and engagement in daily activities, reinforcing the intervention’s broader impact beyond physical growth. In India, starch-heavy diets with limited protein disproportionately affect low-income communities [41]. Therefore, in Indian urban slums, where economic constraints and dietary patterns limit access to animal-sourced foods, the fish waste protein supplementation program is able to directly combat amino acid deficiencies and child growth impairment, offering a scalable and culturally adaptable intervention to tackle child malnutrition effectively.

In the current study, the results demonstrated that APP could restore body weight and height, reduce stunting, and promote muscle mass gain, as evidenced by mid-upper arm circumference (MUAC) improvements—a key indicator of nutritional recovery [42]. The result is consistent with findings from Guinea-Bissau, where a three-month supplementation with additional dairy protein significantly improved weight gain and MUAC, particularly among malnourished children under 59 months of age [43]. Furthermore, this APP study indicates that both the experimental and control groups showed improvements in anemia, a prevalent coexisting condition with child malnutrition in India [44]. These findings align with a randomized controlled trial in India, demonstrating that daily supplementation with 20 g of dairy protein improved weight, height, and anemia outcomes [45]. Future protein supplementation programs should incorporate strategies to address iron and zinc deficiencies in the target community, ensuring a more comprehensive nutritional approach for anemic and malnourished children in India.

However, the BMD findings present a complex picture. Scholars have pointed out that while adequate protein intake supports bone tissue maintenance, insulin-like growth factor 1 stimulation for bone growth, and increased calcium absorption, the metabolism of dietary sulfur amino acids, predominantly from animal protein, may elevate physio-logical acidity, potentially harming long-term bone health [46]. The initial decrease of BMD in subjects was noteworthy, but the long-term effects of bone metabolism warrant further investigation. Additionally, according to the baseline measurement, the high prevalence of calcium and vitamin D deficiency and relatively low lysine intakes were also investigated among stunting participants. Aggarwal and Brains (2022) suggested that dietary and medicinal supplementation of lysine, calcium, and vitamin D may improve the body composition among young participants in the form of proportionally more muscle and bone mass [47]. This recommendation for multiple-nutrient approaches should be considered for further intervention in Indian communities that rely on cereal-based meals with limited lysine and micronutrient content.

Beyond physical growth, childhood malnutrition is strongly linked to cognitive impairment [48]. Caregivers reported improved concentration, self-confidence, and academic performance, attributing these changes to structured feeding and nutrient-rich meals. Prior studies indicate that protein-rich supplementation enhances memory, cerebral blood flow, and lean tissue mass in undernourished children [49]. While no inter-group cognitive differences were detected, intra-group analysis revealed significant cognitive improvements in the APP group, including visual processing, short-term memory, and planning ability. The structured feeding approach fostered by the SCOPE model further contributed to improved social interaction, engagement in learning, and consistent nutrient intake, reinforcing its broader developmental benefits. These mixed-methods findings highlight the additional cognitive advantages of fish waste protein supplementation within the SCOPE model. This makes it a high-impact intervention for addressing malnutrition’s long-term consequences on child development.

The SCOPE framework serves as a transformative, community-driven framework, integrating scientific research, global collaboration, stakeholder engagement, and capacity building to optimize the use of fish waste for malnutrition interventions in the urban slum settings. By leveraging local food resources and industry byproducts, the model ensures accessibility, affordability, sustainability, community ownership, and long-term program success, aligning with the guidance of the Food and Agriculture Organization [50]. This approach addresses immediate nutritional deficiencies, strengthens local food systems, empowers community-based organizations, and fosters social engagement. In this study, SCOPE’s structured, participatory framework was key to ensuring local trust, cultural acceptability, and maintaining high adherence, even in the face of logistical constraints during the COVID-19 pandemic. The integration of culturally familiar meals and responsive caregiver engagement highlights the importance of embedding nutritional interventions within specific contexts of target communities.

From a policy standpoint, the findings underscore the potential of APP as an innovative supplement within public nutrition platforms. Policymakers may consider integrating sustainable proteins like APP into existing school feeding programs or community-based child nutrition schemes, especially in protein-insecure urban slum settings. Such integration could expend reach while supporting local economies through the use of underutilized fish bycatch. Despite external challenges, particularly the COVID-19 pandemic, the study provides compelling evidence for the global scalability of APP supplementation. Future research should explore adaptation across diverse cultural and food system contexts, reinforcing the need for community-driven, sustainable solutions to combat protein malnutrition and food insecurity worldwide. The SCOPE model offers a replicable framework for long-term nutritional resilience in vulnerable populations by integrating culturally appropriate feeding, nutritional education, and real-time monitoring.

Several limitations must be acknowledged. First, the final sample size (*n* = 46) was relatively small due to COVID-19-related disruptions and household-level constraints, which may have reduced the statistical power and limited the external validity of the findings. As a result, we were unable to perform age-stratified analyses, although children aged 3–6 may exhibit differential responsiveness to protein supplementation. Second, the study population was recruited from the local health framework in a single urban slum community in Bengaluru, potentially limiting generalizability to other geographic or cultural settings. Third, the 90-day intervention period with 180-day follow-up allowed for short-term evaluation of physical and cognitive outcomes but was not sufficient to assess long-term developmental or nutritional effects. Additionally, the qualitative component relied solely on focus group discussions with caregivers, whose perspectives, though valuable, may have been influenced by recall limitations or social desirability bias. This caregiver-only lens may not fully capture the broader programmatic or systemic challenges involved in implementing community-based protein supplementation. To strengthen triangulation and gain a more comprehensive understanding of implementation feasibility and barriers, future studies should include in-depth interviews with frontline health workers, childcare staff, and local stakeholders such as community leaders and nutrition officers. Fifth, unmeasured confounders—such as seasonal variation in food availability, child physical activity, or infection status—were not accounted for and could have influenced outcomes. Finally, ethnicity, geography, and cultural dietary practices may affect both the acceptability and physiological impact of protein supplementation. These factors were not controlled in the current study but should be explored in future multicenter research to assess heterogeneity in response. Future research should address these limitations through larger, multisite trials with extended follow-up and objective developmental assessments to evaluate the long-term efficacy and scalability of APP supplementation across diverse settings.

## 5. Conclusions

Through the Sustainable Community Partnership and Empowerment (SCOPE) strategy, the community-based mixed-methods study employing an ocean-based protein (APP) fortification approach demonstrated improvements in the weight, height, mid-upper arm circumference, and cognition of malnourished children in urban slums. Caregivers’ positive perspectives further affirmed child behavior change and the cultural acceptability and feasibility of this community-based approach. This research substantiates the potential for the fish waste supplement to be an innovative, affordable, culturally sensitive, and sustainable solution to child malnutrition in urban slums of low- and middle-income nations. Policymakers may consider integrating the sustainable protein into school feeding programs or community-based nutrition schemes, particularly in protein-insecure urban settings. Future research should involve larger, multi-site samples and longitudinal designs to evaluate long-term growth, cognitive, and behavioral outcomes across diverse socio-cultural contexts.

## Figures and Tables

**Figure 1 nutrients-17-01751-f001:**
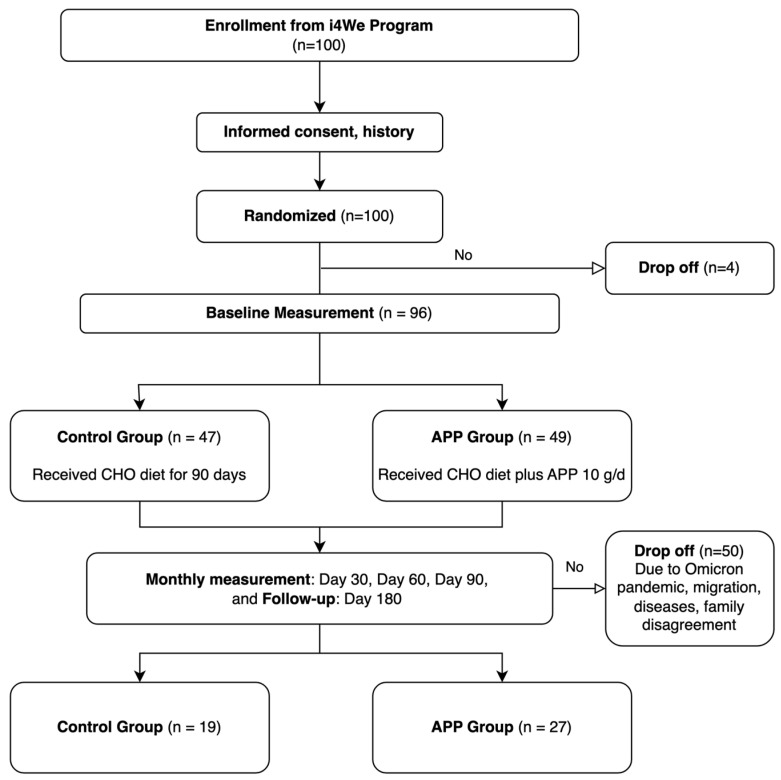
Research flow.

**Figure 2 nutrients-17-01751-f002:**
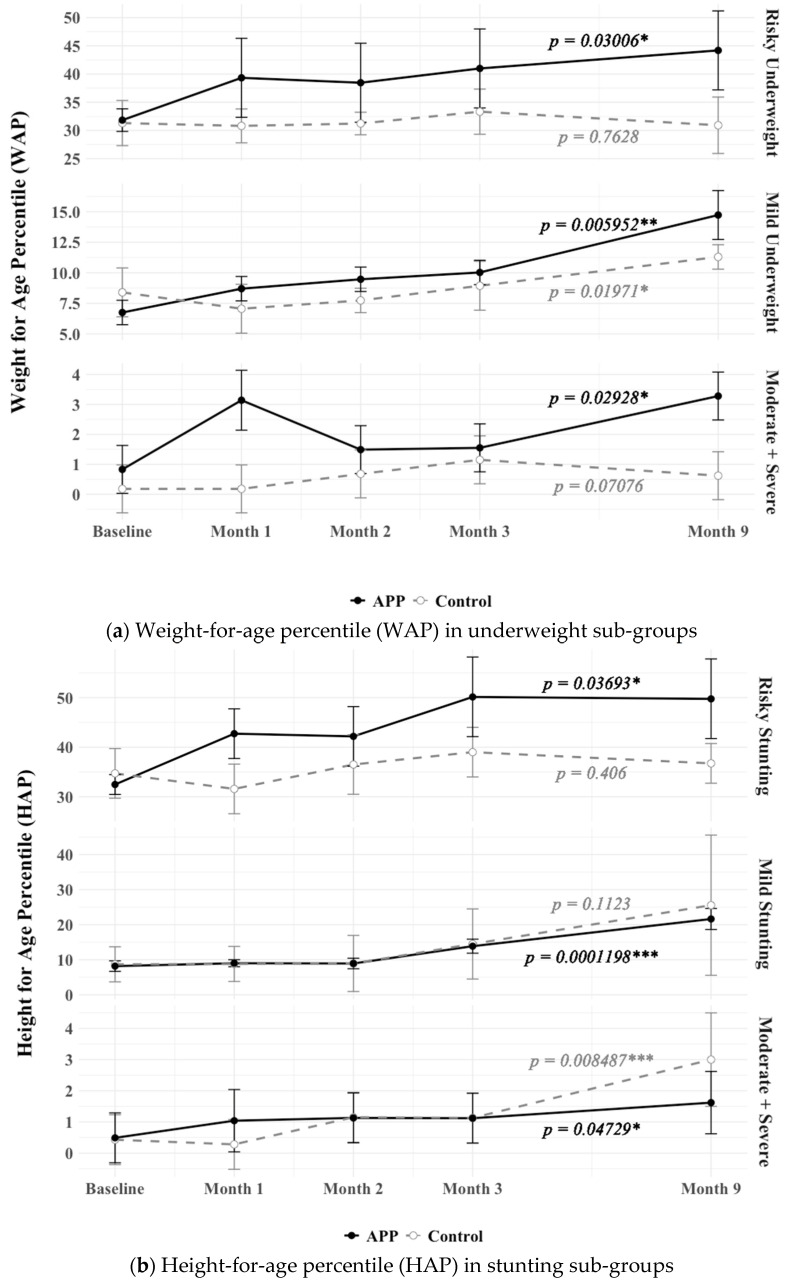
The effects of APP supplementation on (**a**) weight-for-age percentile (WAP) among multiple levels of underweight, and (**b**) height-for-age percentile (HAP) among multiple levels of stunting. Data are presented as mean ± SEM. Statistical analysis was performed using the Friedman test to evaluate differences across time points. Significant differences were indicated by *p*-values < 0.05 *, < 0.01 **, and < 0.001 ***.

**Figure 3 nutrients-17-01751-f003:**
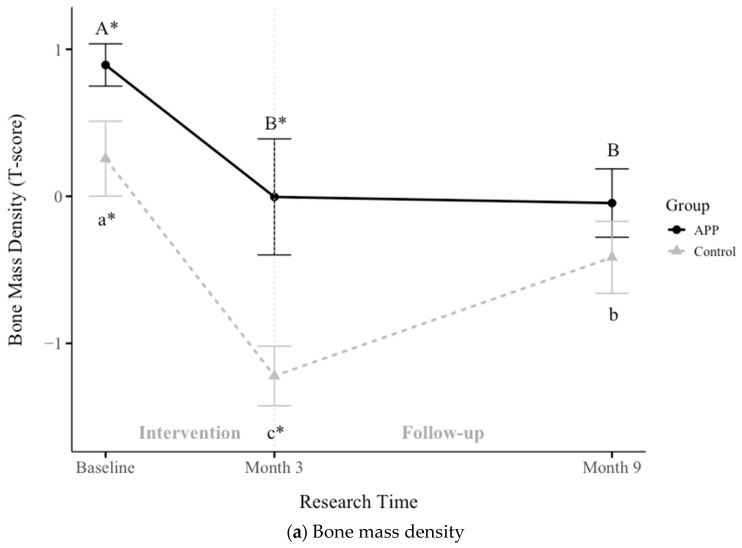
The effects of APP supplementation on (**a**) bone mass density, (**b**) middle-upper arm circumstance (MUAC), and (**c**) hemoglobin levels. Data are presented as mean ± SEM. Statistical analysis was performed using the nonparametric Wilcoxon test to compare differences between the two groups, with significant differences indicated by *. The Friedman test was used to evaluate differences across time points, with significant differences denoted by capital letters in the APP group and lowercase letters in the Control group. A *p*-value < 0.05 was considered statistically significant.

**Figure 4 nutrients-17-01751-f004:**
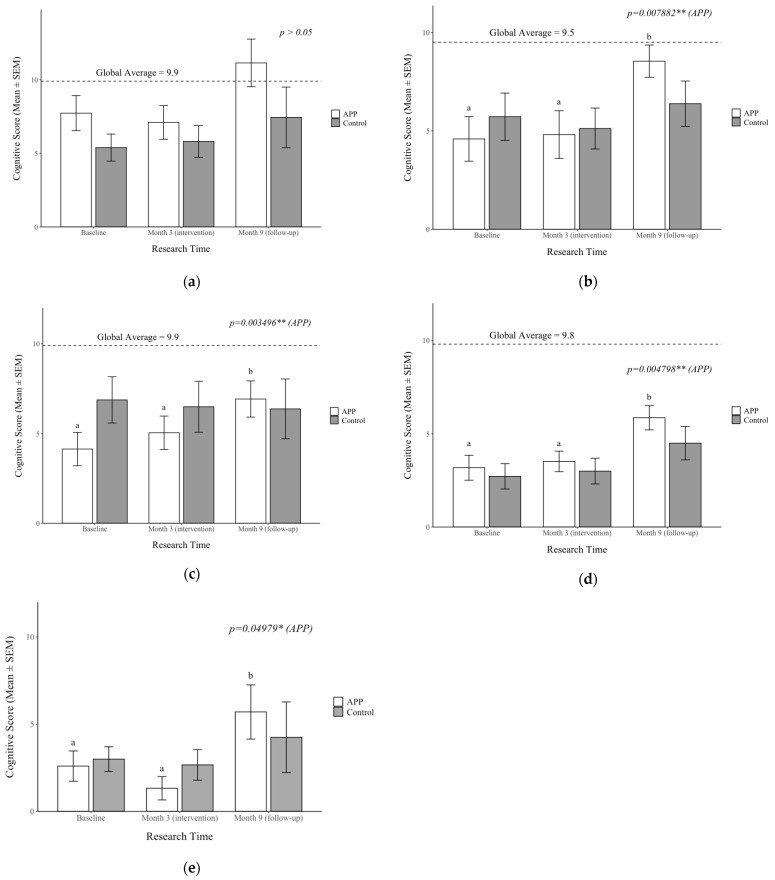
The effects of APP supplementation on cognition: (**a**) conceptual thinking; (**b**) visual processing as assessed by the triangles test; (**c**) visual processing as assessed by the face recognition test; (**d**) short-term memory as assessed by the hand movement test; (**e**) planning ability as assessed by the pattern reasoning test. Data are presented as mean ± SEM. Statistical analysis was performed using the Friedman test to evaluate differences across time points, with significant differences indicated by letters. Significant differences were indicated by *p* < 0.05 *, and <0.01 **.

**Table 1 nutrients-17-01751-t001:** Baseline measurement of subjects ^1,2,3^.

Indicators	Total	Control	APP	*p*-Value
	(*n* = 46)	(*n* = 19)	(*n* = 27)	
Age (year) ^1^	4.31 ± 0.11	4.6 ± 0.28	4 ± 0.2	0.05792
Gender (Boy/Girl, *n*) ^2^	28:20	12:7	14:13	0.6458
Growth indicators				
Weight (kg) ^1^	14.77 ± 0.61	15.03 ± 0.68	14.58 ± 0.94	0.1291
WAP ^1^	24.97 ± 4.1	25.18 ± 5.92	24.83 ± 5.71	0.9644
Underweight (*n*, %) ^2^	12 (26.09)	4 (21.05)	8 (29.62)	0.7556
Height (cm) ^1^	97.55 ± 1.41	100.03 ± 2.16	95.8 ± 1.8	0.1132
HAP ^1^	19.82 ± 3.58	24.78 ± 6.9	16.34 ± 3.66	0.7037
Stunting (*n*, %) ^2^	18 (39.13)	8 (42.1)	10 (37.03)	0.9681
MUAC (cm) ^1^	15.97 ± 0.3	15.88 ± 0.27	16.03 ± 0.48	0.2323
BMD (T-score) ^1^	0.63 ± 0.14	0.26 ± 0.25	0.89 ± 0.14	0.03903 *
Hemoglobin (%) ^1^	9.17 ± 0.34	9.29 ± 0.41	9.09 ± 0.51	0.8759
Dietary intake				
Energy (Kcal/day) ^1^	908.38 ± 98.64	878.32 ± 65.26	929.53 ± 47.81	0.4129
Carbohydrate (g/d) ^1^	116.22 ± 6.02	112.95 ± 9.89	118.52 ± 7.67	0.5506
Protein (g/d) ^1^	21.16 ± 1.05	19.57 ± 1.41	22.28 ± 1.41	0.2676
Fat (g/d) ^1^	35.11 ± 1.79	33.84 ± 3.26	36 ± 2.05	0.3759

^1^ Data represent mean ± SEM, Wilcoxon Test used for group difference. Significance levels are indicated as follows: *p*-value < 0.05 *. ^2^ Data represent sample number and percentage. Statistical analysis was performed using the chi-squared test (*p* < 0.05). ^3^ Weight-for-age percentile, WAP; Height-for-age percentile, HAP; Mid-upper arm circumference, MUAC; Bone mass density, BMD.

**Table 2 nutrients-17-01751-t002:** The progress of body weight (kg) ^1,2^.

	Baseline	Intervention			Follow up	*p*-Value
	Month 0	Month 1	Month 2	Month 3	Month 9	
Control (*n* = 19)	15.03 ± 0.68 ^a^	15.31 ± 0.7 ^b^	15.63 ± 0.71 ^c^	15.75 ± 0.67 ^c^	16.47 ± 0.76 ^d^	<0.001 ***
Non-underweight (*n* = 3)	18.2 ± 0.7	18.67 ± 1.13	18.9 ± 1.3	18.63 ± 0.99	20.13 ± 1.72	0.406
Risky underweight (*n* = 7)	16.46 ± 0.94 ^a^	16.74 ± 0.94 ^a^	17.1 ± 0.99 ^ab^	17.3 ± 0.95 b	17.3 ± 1.29 ^ab^	0.008466 **
Mild underweight (*n* = 5)	14.1 ± 0.71 ^a^	14.36 ± 0.63 ^a^	14.48 ± 0.66 ^a^	14.74 ± 0.69 ^a^	15.88 ± 0.72 ^b^	0.001222 **
Moderate + severe underweight (*n* = 4)	11.33 ± 0.69 ^a^	11.48 ± 0.68 ^a^	12.05 ± 0.66 ^a^	12.15 ± 0.49 ^a^	13 ± 0.47 ^b^	0.008569 **
APP (*n* = 27)	14.58 ± 0.94 ^a^	15.28 ± 0.95 ^b^	15.42 ± 0.93 ^bc^	15.49 ± 0.97 ^c^	16.06 ± 1.01 ^d^	<0.001 ***
Non-underweight (*n* = 4)	24.2 ± 3.16	25.05 ± 2.84	25.3 ± 2.38	25.15 ± 3.1	25.48 ± 3.69	0.3007
Riskily underweight (*n* = 9)	14.24 ± 0.26 ^a^	15.1 ± 0.5 ^b^	15.26 ± 0.53 ^bc^	15.57 ± 0.52 ^c^	16.5 ± 0.54 ^d^	<0.001 ***
Mildly underweight (*n* = 6)	13.13 ± 0.49 ^a^	13.78 ± 0.42 ^b^	13.65 ± 0.35 ^bc^	13.98 ± 0.49 ^bc^	13.68 ± 1.12 ^bc^	0.01083 *
Moderately + severely underweight (*n* = 8)	11.24 ± 0.44 ^a^	11.78 ± 0.55 ^b^	12 ± 0.56 ^b^	11.81 ± 0.51 ^b^	12.8 ± 0.53 ^c^	<0.001 ***

^1^ Data are presented as mean ± SEM. The underweight subgroups in the APP and Control groups are categorized based on the weight-for-age Z-score (WAZ): non-underweight (WAZ > 0), risky (−1 < WAZ < 0), mildly underweight (−2 < WAZ ≤ −1), and moderately + severely underweight (WAZ ≤ −2). ^2^ The Friedman test was used to evaluate differences across time points, with significant differences indicated by *p*-values < 0.05 *, <0.01 **, and <0.001 ***, which were considered statistically significant. Post hoc Wilcoxon tests were performed to compare two different time points, with significant statistical differences denoted by letters when *p* < 0.05.

**Table 3 nutrients-17-01751-t003:** The progress of body height (cm) ^1,2^.

	Baseline	Intervention			Follow up	*p*-Value
	Month 0	Month 1	Month 2	Month 3	Month 9	
Control (*n* = 19)	101.04 ± 2.36 ^a^	101.19 ± 2.11 ^a^	102.16 ± 2.11 ^b^	103.25 ± 2.14 ^c^	107.04 ± 2.06 ^d^	<0.0001 ***
Non-stunted (*n* = 3)	107 ± 2.52 ^a^	108.23 ± 3.27 ^a^	109.1 ± 2.52 ^a^	109.83 ± 2.12 ^a^	114.33 ± 3.01 ^b^	0.04477 *
Risky stunting (*n* = 6)	105.22 ± 2.42 ^a^	106.4 ± 2.29 ^a^	107.43 ± 2.09 ^a^	108.32 ± 2.18 ^ab^	111.12 ± 2.3 ^b^	0.000146 *
Mild stunting (*n* = 2)	103.75 ± 14.25	104.7 ± 13.3	106.4 ± 14.1	107.5 ± 14.7	110.85 ± 13.95	0.0954
Moderate + Severe (*n* = 8)	95 ± 3.69 ^ab^	93.76 ± 1.76 ^a^	94.54 ± 1.64 ^a^	95.93 ± 1.97 ^b^	100.31 ± 1.87 ^c^	<0.0001 ***
APP (*n* = 27)	95.8 ± 1.81 ^a^	97.66 ± 1.84 ^b^	97.97 ± 1.86 ^b^	99.56 ± 1.88 ^c^	103.03 ± 1.8 ^d^	<0.0001 ***
Non-stunted (*n* = 2)	115.45 ± 3.45	117.25 ± 5.25	117.75 ± 4.75	118.45 ± 5.05	120.8 ± 4.6	0.1039
Risky stunting (*n* = 8)	100.84 ± 2.42 ^a^	103.08 ± 2.14 ^b^	103.28 ± 2.31 ^b^	105.88 ± 2.13 ^c^	109. 63 ± 1.92 ^d^	<0.0001 ***
Mild stunting (*n* = 2)	94.34 ± 2 ^a^	96.03 ± 2.26 ^b^	96.27 ± 2.34 ^b^	97.87 ± 2.21 ^c^	101.1 ± 2.22 ^d^	<0.0001 ***
Moderate + Severe (*n* = 10)	88. 87 ± 1.83 ^a^	90.56 ± 1.92 ^b^	90.95 ± 1.64 ^b^	91.9 ±1.89 ^c^	95.54 ± 1.73 ^d^	<0.0001 ***

^1^ Data are presented as mean ± SEM. The stunting subgroups between the APP and Control groups are categorized based on the height-for-age Z-score (HAZ): non-stunted (HAZ > 0), risky (−1 < HAZ < 0), mild stunting (−2 < HAZ ≤ −1), and moderate + severe stunting (HAZ ≤ −2). ^2^ The Friedman test was used to evaluate differences across time points, with significant differences indicated by *p*-values < 0.05 *, and <0.001 ***, which were considered statistically significant. Post hoc Wilcoxon tests were performed to compare two different time points, with significant differences denoted by letters when *p* < 0.05.

## Data Availability

The data presented in this study are available on request from the corresponding author due to ethical considerations and confidentiality agreements protecting the privacy of child participants and their families in vulnerable communities.

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
