# Peer review of "A Community-Based Mixed-Methods Study: Fish Bycatch Protein Supplementation as a Sustainable Solution for Child Malnutrition in Bengaluru, India"

_nutrients, 2025, doi:10.3390/nu17111751_

Round 1
Reviewer 1 Report
Comments and Suggestions for Authors
The work carried out by Yang et al can be considered for publication in Nutrients after some adjustments. These are my suggestions:
The study’s aims need to be better clarified in the abstract. In this section, the main results should be expressed quantitatively, and some directions to be taken in future investigations should be indicated. It also exceeds the 250-word limit.
A more robust background has to be provided in the introductory section. In its current state, it is too short and brief. A good justification for carrying out this study has to be given, and the addressed topics can be deeper analyzed.
Mention the approval date provided by the Ethics Committee for your study.
In the Materials and Methods, please justify the adequacy of your sample size. It seems a small sample and not representative of the study population.
The Results are well described and organized. However, I can’t say the same regarding the Discussion. Please, reorganize it in a better way by dividing it into subsections, aligned with the Results; Include more citations of other similar research carried out in other countries and continents, and compare those studies with your obtained results; Mention and discuss your study’s strengths and limitations.
The Conclusions need to be improved as well. Elaborate on the practical and policy implications of your study, its impact on the population and scientific community, and suggest some directions for further studies.
Author Response
Thank you very much for reviewing our manuscript. Please find our point-by-point responses below, along with the revised version of the manuscript with track changes attached.
Comments 1: The study’s aims need to be better clarified in the abstract. In this section, the main results should be expressed quantitatively, and some directions to be taken in future investigations should be indicated. It also exceeds the 250-word limit.
Response 1: Thank you for this valuable suggestion. We have revised the abstract to clarify the study’s aims, present the main results with quantitative data, and indicate directions for future research. The revised abstract (lines 11–32, page 1) now adheres to the 250-word limit. Major revisions are highlighted in red for clarity.
Revised abstract:
Objective: Malnutrition remains a global challenge to child development, with urban slums in India experiencing high rates of protein deficiency. This study aimed to evaluate the effectiveness of a low-cost, fish bycatch-derived protein supplement in supporting catch-up growth among malnourished children.Methods: Using the Sustainable Community Partnership and Empowerment (SCOPE) model, we implemented a 90-day intervention with daily 10 grams of Advanced Protein Powder (APP), produced from underutilized fish bycatch. Forty-six malnourished children (aged 3–6) from Bengaluru slums were randomized into a Control group receiving caloric support or an APP supplement group. Growth indicators, cognition, and caregiver perspectives were assessed. Results: Children receiving 90-day APP showed a significant increase in the Weight-for-Age Percentile (underweight), rising by 7.59 %, compared to 0.59 % in the Control group (p = .02185). Muscle growth, measured by mid-upper arm circumference, also improved significantly in the APP Group (p < .05). In the first month, APP supplementation led to a significant height gain of 1.86 cm (p < .001), whereas the Control group showed no change (p > .05). Additionally, APP supplementation enhanced cognitive function-visual processing, short-term memory and planning-with sustained effects after six months (p < .05). Caregivers reported noticeable improvements in children’s vitality, appetite, focus, and engagement in social and learning activities. Conclusions: Bycatch-derived protein supplementation, implemented through the SCOPE model, enhanced physical growth, behavior, and cognition in malnourished children in urban slums. Future studies should investigate the long-term effects, scalability, and adaptability of this sustainable solution for addressing child malnutrition. (250 words)
Comments 2: A more robust background has to be provided in the introductory section. In its current state, it is too short and brief. A good justification for carrying out this study has to be given, and the addressed topics can be deeper analyzed.
Response 2:
Thank you for this helpful comment. We have thoroughly revised the Introduction section to provide a more robust background. Specifically, we expanded three parts:
• First, we strengthened the justification for community-based interventions in the urban slums of Bengaluru, as reflected in lines 45–56 on page 2.
• Second, we revised the middle paragraph (lines 57–75 on page 2) to better articulate the rationale for utilizing fish bycatch-derived protein, emphasizing its cultural acceptability, sustainability, and affordability in urban slum settings.
• Third, we added references to previous studies on this protein source and included a new description on directions for future research in the final paragraph of the introduction (lines 78 on page 2 to 94 on page 3).
These changes aim to enhance the clarity, depth, and relevance of the study’s rationale. Please check the text in attached file with tracking change.
Comments 3: Mention the approval date provided by the Ethics Committee for your study.
Response 3: Thank you for this suggestion. We have revised the manuscript accordingly. The ethics approval dates have been added to the Methods section on Page 3, Lines 111–112.
Updated text: The Institutional Review Boards of Oregon State University (Oregon, USA; IRB-2019-0330, approved on November 21, 2019) and the Catalyst Foundation (Bengaluru, India; approved on August 26, 2019) approved the study protocol.
Comments 4: In the Materials and Methods, please justify the adequacy of your sample size. It seems a small sample and not representative of the study population.
Response 4: Thank you for pointing this out. We this comment. Accordingly, we have added information regarding the household representation rate—from the initial 100 participants to the final 46—on Page 3, Lines 97–100 and 115–116, to clarify the sampling rationale and constraints. Although the final sample size may appear small, it reflects real-world challenges during the COVID-19 pandemic, including household instability, safety concerns, and limited access for continued participation. Despite this, the sample represents 11.44% of the total households in the target slum population, and the study context remains clearly defined and demographically concentrated, supporting the relevance of the findings to similar high-risk urban environments.
Updated text:
This study initially recruited 100 children aged 3–6 years and their primary caregivers living in the urban slums in Bommanahalli, Bengaluru, India, from November 2021 to December 2022, representing approximately 24.87 % of the 402 households in the targeted community during the preliminary screening. Conducted in collaboration with Swasti, a non-profit organization supporting marginalized communities and low-income families, these child-parent participants were selected from the i4We program, Swasti’s integrated healthcare framework. Although the sampling strategy was a limitation for broader generalizability, the clearly defined and demographically concentrated setting enhances the validity of the findings for similar high-risk urban contexts.
The final sample included 19 participants in the Control group and 27 in the APP group, representing 11.44% of total household in the target population.
Comments 5: The Results are well described and organized. However, I can’t say the same regarding the Discussion. Please, reorganize it in a better way by dividing it into subsections, aligned with the Results; Include more citations of other similar research carried out in other countries and continents, and compare those studies with your obtained results; Mention and discuss your study’s strengths and limitations.
Response 5: Thank you for this valuable feedback. We have thoroughly revised the Discussion section to enhance clarity and alignment with the Results. The revised section begins with a general overview of the study’s significance, followed by clearly organized subsections that correspond to the major outcome domains: dietary intake, growth impact, bone and hemoglobin, and cognitive development. Relevant qualitative findings from caregivers have been integrated within each corresponding subsection to provide contextual insight and reinforce the quantitative results. Additionally, we have expanded the discussion by incorporating citations from studies conducted in other countries, enabling cross-contextual comparison with our findings in second paragraph. Moreover, a new paragraph outlining policy recommendations has also been included, highlighting the practical implications of our intervention. Lastly, we have added a comprehensive paragraph discussing the study’s limitations. These changes can be found in the revised Discussion section from page 15 to page 18, lines 450 to 579.
Comments 6: The Conclusions need to be improved as well. Elaborate on the practical and policy implications of your study, its impact on the population and scientific community, and suggest some directions for further studies.
Response 6: Thank you for the conclusion suggestion. Accordingly, we have revised the conclusion section on Page 18, Lines 580–593 to expand on the practical, policy, and scientific implications of our findings and propose directions for future research.
Updated text:
Through the Sustainable Community Partnership and Empowerment (SCOPE) strategy, the community-based mixed-methods study employing an ocean-based protein (APP) fortification approach demonstrated improvements in the weight, height, mid-upper arm circumference, and cognition of malnourished children in urban slums. Caregivers’ positive perspectives further affirmed child behavior change and the cultural acceptability and feasibility of this community-based approach. This research substantiates the potential for the fish waste supplement to be an innovative, affordable, culturally sensitive, and sustainable solution to child malnutrition in urban slums of low- and middle-income nations. Policymakers may consider integrating the sustainable protein into school feeding programs or community-based nutrition schemes, particularly in protein-insecure urban settings. Future research should involve larger, multi-site samples and longitudinal designs to evaluate long-term growth, cognitive, and behavioral out-comes across diverse socio-cultural contexts.

Reviewer 2 Report
Comments and Suggestions for Authors
Report 1
Hello, thank you for the opportunity to evaluate this article.
I believe that the article is original, and the information is presented in a clear manner, respecting scientific rigor.
I would like to make a few suggestions and also to ask some questions to the authors:
1.Please mention the limitations of your study.
2.Do you think that this type of study could be extended to other categories of patients in the future?
3. You mentioned a similar study from Guinea-Bissau. What would be the particularities noted in the patients in this study compared to that study or other similar studies?
4. Could ethnicity and geography influence the effect? ​​Have you noticed other factors that could have influenced the response to the APP diet?
5. Were there differences in response to the APPC diet in different age groups? If so, what could be the causes?
Thank you.

Author Response
Thank you very much for your thoughtful review of our manuscript. We sincerely appreciate your time and valuable feedback. Please find our detailed point-by-point responses to each of your comments below, along with the revised manuscript with tracked changes attached for your reference.
Comments1: Please mention the limitations of your study.
Response 1: Thank you for pointing this out. We have added a comprehensive paragraph discussing the study’s limitations. These changes can be found in the revised Discussion section on page 18, lines 554 to 579.
Updated text:
Several limitations must be acknowledged. First, the final sample size (n = 46) was relatively small due to COVID-19-related disruptions and household-level constraints, which may have reduced the statistical power and limited the external validity of the findings. As a result, we were unable to perform age-stratified analyses, although chil-dren aged 3–6 may exhibit differential responsiveness to protein supplementation. Second, the study population was recruited from the local health framework in a single urban slum community in Bengaluru, potentially limiting generalizability to other geographic or cultural settings. Third, the 90-day intervention period with 180-day fol-low-up allowed for short-term evaluation of physical and cognitive outcomes but was not sufficient to assess long-term developmental or nutritional effects. Additionally, the qualitative component relied solely on focus group discussions with caregivers, whose perspectives, though valuable, may have been influenced by recall limitations or social desirability bias. This caregiver-only lens may not fully capture the broader program-matic or systemic challenges involved in implementing community-based protein sup-plementation. To strengthen triangulation and gain a more comprehensive under-standing of implementation feasibility and barriers, future studies should include in-depth interviews with frontline health workers, childcare staff, and local stakeholders such as community leaders and nutrition officers. Fifth, unmeasured confounders—such as seasonal variation in food availability, child physical activity, or infection status—were not accounted for and could have influenced outcomes. Finally, ethnicity, geography, and cultural dietary practices may affect both the acceptability and physiological impact of protein supplementation. These factors were not controlled in the current study but should be explored in future multicenter research to assess heterogeneity in response. Future research should address these limitations through larger, multisite trials with extended follow-up and objective developmental assessments to evaluate the long-term efficacy and scalability of APP supplementation across diverse settings.
Comments 2:Do you think that this type of study could be extended to other categories of patients in the future?
Response 2: Thank you for this insightful question. We believe that this type of study can indeed be extended to other populations and patient categories across diverse settings. The community-based approach, implemented through the Sustainable Community Partnership and Empowerment (SCOPE) model, emphasizes cultural relevance, local ownership, and collaboration with local partners. These features make the intervention adaptable to various socio-cultural contexts, provided that policymakers and healthcare practitioners are committed to understanding community needs, respecting local perspectives, and working collaboratively.
To address this point, we have added a paragraph in the discussion (Page17, Line 549–553):
Future research should explore adaptation across diverse cultural and food system contexts, reinforcing the need for community-driven, sustainable solutions to combat protein malnutrition and food insecurity worldwide. The SCOPE model offers a replicable framework for long-term nutritional resilience in vulnerable populations by integrating culturally appropriate feeding, nutritional education, and real-time monitoring.
Comment 3: You mentioned a similar study from Guinea-Bissau. What would be the particularities noted in the patients in this study compared to that study or other similar studies?
Response 3: Thank you for pointing this out. We have clarified the comparison between our study and the Guinea-Bissau intervention in the discussion section [Page 16, Lines 491–494].
Updated text: The result is consistent with findings from Guinea-Bissau, where a three-month supplementation with additional dairy protein significantly improved weight gain and MUAC, particularly among malnourished children under 59 months of age.
Comments 4: Could ethnicity and geography influence the effect? ​​Have you noticed other factors that could have influenced the response to the APP diet?
Response 4: Thank you for this thoughtful comment. We agree that ethnicity and geography may influence the effects of nutritional interventions. While our study population was relatively homogenous—drawn from a single urban slum community in Bengaluru with shared socioeconomic and dietary backgrounds—we recognize that geographic location, regional food systems, cultural norms, and ethnic dietary preferences could all influence the acceptability, bioavailability, and metabolic response to the APP supplementation. Additionally, other unmeasured factors such as baseline gut health, habitual diet, infection burden, and household-level food sharing practices may have influenced the outcomes. We have added this point to the limitation section [Page 18, Lines 571–576], noting that future studies should stratify by ethnicity, geographic context, and diet diversity to better assess variability in response to fish protein supplementation across diverse populations.
Updated text:
Fifth, unmeasured confounders—such as seasonal variation in food availability, child physical activity, or infection status—were not accounted for and could have influenced outcomes. Finally, ethnicity, geography, and cultural dietary practices may affect both the acceptability and physiological impact of protein supplementation. These factors were not controlled in the current study but should be explored in future multicenter research to assess heterogeneity in response.
Comments 5: Were there differences in response to the APP diet in different age groups? If so, what could be the causes?
Response 5: Thank you for this insightful question. In our analysis, children were aged between 3 and 6 years, and although age was recorded and baseline age distribution was statistically similar between the APP and Control groups, the study was not powered to detect age-stratified differences in response to the intervention. Therefore, we did not perform subgroup analyses by narrower age brackets. However, it is plausible that younger children may exhibit greater responsiveness to protein supplementation due to their higher relative growth velocity and nutritional needs. We have now added a brief mention of this point in the discussion section (Page 18, Line 557–558), suggesting that future research should examine age-specific responses to APP supplementation using larger sample sizes and stratified analysis to better understand differential growth outcomes by developmental stage.
Updated text: As a result, we were unable to perform age-stratified analyses, although children aged 3–6 may exhibit differential responsiveness to protein supplementation.
